# OpenReview forum: "Can Language Models Reason about Individualistic Human Values and Preferences?"
_ICLR.cc/2025/Conference — ICLR 2025 Conference Withdrawn Submission_

### Official Review · Reviewer_4q73 · 2024-10-25

**Soundness:** 2
**Presentation:** 3
**Contribution:** 2
**Rating:** 3
**Confidence:** 2

**Summary:**

This paper introduces the concept of "individualistic alignment" to capture human values of individuals. This paper presents the INDIEVALUECATALOG, a dataset derived from the World Values Survey, which includes standardized statements expressing individual preferences from a global sample of individuals. A novel metric, the Value Inequity Index (σINEQUITY), is proposed to assess the impartiality of models across demographics.

**Strengths:**

(1) The study of prediction equity across demographic groups is interesting, and the result is insightful.

(2) A new metric, the Value Inequity Index (σINEQUITY), is proposed to measure how equitably models treat different demographics.

(3) This paper tested multiple LLMs.

**Weaknesses:**

(1) The problem formalization with notations in Section 2.2 is unnecessarily complicated.

(2) The focus on predicting individual values may be problematic. Even individuals with the same demographics can differ widely due to various factors. This means the data contains a lot of randomness and noises. This could be why the models struggled to perform well, even after fine-tuning on similar data. A group/demographic-level setting might be more reasonable.

(3) Why the studied task is important and useful in real-world applications needs further explanation.

(4) The technical innovation is limited. The core contribution, the INDIEVALUECATALOG dataset, is essentially a simple conversion of the World Values Survey. The authors need to explain more about the core novelty of this paper.

(5) Data and code is not currently shared.

**Questions:**

Considering the inherent variability and potential noise and randomness in predicting individual values, could you elaborate on the real-world applications where this task would be particularly impactful? How do you envision these predictions being useful in practical scenarios?

---

### Official Review · Reviewer_rvnN · 2024-10-31

**Soundness:** 3
**Presentation:** 3
**Contribution:** 3
**Rating:** 6
**Confidence:** 4

**Summary:**

This paper addresses the limitations of previous pluralistic alignment approaches that pre-categorize individuals, highlighting the importance of "individualistic alignment". To achieve "individualistic alignment", the authors introduce the INDIEVALUECATALOG dataset based on the World Values Survey (WVS). Through experimental validation, they reveal the limited capability of current state-of-the-art LLMs in understanding individualistic human values, as measured by the Value Inequity Index (σINEQUITY) proposed by authors. Furthermore, the authors train a collection of Individualistic Value Reasoners (INDIEVALUEREASONER) models on INDIEVALUECATALOG, enhancing LLMs' capabilities in individualistic value reasoning.

**Strengths:**

* This paper focuses on individualistic alignment, which is an interesting and novel topic
* This paper utilizes World Values Survey (WVS) data and transforms it into a format suitable for LLM training, creating the INDIEVALUECATALOG dataset
* This paper proposes the VALUE INEQUITY INDEX (σINEQUITY) to measure the fairness of model reasoning across different demographic groups, revealing the current limitations of SOTA LLMs in this aspect
* Through fine-tuning LLMs on INDIEVALUECATALOG, authors explore how to combine proposed dataset and LLMs to discover patterns in human values

**Weaknesses:**

* Despite the interesting problem setting, the technical contributions of this paper appear limited for ICLR.

* There is insufficient discussion of the practical application value of "individualistic alignment"

* The paper lacks performance comparisons with related work

**Questions:**

* While using human preference data for general value alignment has significantly improved LLMs' capabilities as assistants, does this paper's proposed "individualistic alignment" offer further performance improvements or broader application value?

* Are advanced pluralistic alignment methods technically comparable with the method proposed in this paper, and could performance comparisons with related work be conducted to highlight the relative effectiveness of the proposed method?

---

### Official Review · Reviewer_wkhE · 2024-11-02

**Soundness:** 2
**Presentation:** 2
**Contribution:** 2
**Rating:** 3
**Confidence:** 4

**Summary:**

This paper investigated the limitations of LLMs in reasoning about human values at an individual level. The main contributions include:
1. Presented a new dataset, INDIEVALUECATALOG, derived from the World Values Survey (WVS) that transforms unstructured survey questions into standardized natural language statements representing value preferences.
2. Examined LLMs' abilities to predict individuals' values based on a set of their value-expressing statements.
3. Trained LLMs with individualistic value statements to achieve proficient individual value reasoners.

**Strengths:**

1. The work tackles a crucial challenge in AI alignment: understanding human values at an individual level rather than relying on broad demographic categories. This bottom-up approach overcomes the limitations of traditional demographic-based models, enabling the development of AI systems that are both more equitable and better tailored to individual needs.
2. The paper's visualization is particularly effective. Figure 1 provides a clear illustration of the author's concept of individualistic value reasoning.
3. The paper conducted a thorough empirical evaluation on demonstrating the proficiency of trained individual value reasoners, including the comparison between various state-of-the-art LLMs and statistical methods.

**Weaknesses:**

1. The methodology lacks novelty. The training of individualistic value reasoner relies solely on fine-tuning approaches; the proposed metrics on LM proficiency and impartiality offer no novel contributions, and the overall methodological approach contains no significant innovations.
2. The paper's analysis lacks sufficient depth and fails to make substantial contributions to the field. While it identifies a key limitation - namely, frontier LLMs' deficiency in understanding and predicting individualistic human values - this observation, though intuitively correct, merely confirms what was already suspected. The paper does not extend beyond this basic insight to provide meaningful scholarly contributions.
3. The paper suffers from disjointed content and lacks coherent logical flow. Section 3 presents various findings about LLMs' accuracy in predicting individualistic values, but these emerge as scattered observations rather than systematic research. While these findings, such as identifying which demographic groups' values are more accurately predicted, are intuitively reasonable, they fail to coalesce into a comprehensive study of the field. The paper ultimately reads as a collection of disparate data analyses lacking meaningful synthesis or substantive theoretical contributions.

**Questions:**

The author's primary task involves predicting an individual's value judgments in novel situations based on a sample of their value-expressing statements. However, two critical questions emerge: First, are these provided value-expressing statements sufficient to capture a person's complete worldview and value system? Second, how can the authors differentiate between prediction inaccuracies caused by incomplete value statements versus those stemming from limitations in the LLM's reasoning capabilities?

---

### Official Review · Reviewer_SviC · 2024-11-02

**Soundness:** 3
**Presentation:** 2
**Contribution:** 3
**Rating:** 6
**Confidence:** 3

**Summary:**

This paper introduces a dataset from the World Values Survey designed to evaluate language models' (LMs) reasoning on individualistic values. Unlike pluralistic alignment approaches that generalize diversity through demographic categories, this dataset supports more nuanced, individual-focused alignment. With 93K participants’ value statements, the study highlights LMs' limitations in predicting individual preferences (accuracy 55-65%) and introduces the VALUE INEQUITY INDEX (σINEQUITY) to assess model impartiality. Trained Individualistic Value Reasoners show slight accuracy improvements, providing new insights into global individual values.

**Strengths:**

1. The proposed dataset is a significant addition, transforming unstructured WVS data into a structured, standardized resource for examining individualistic values. This dataset enables a more granular approach to evaluate the performance of human value reasoning on LLMs.

2. The paper’s critique of pluralistic alignment's reliance on broad demographic categories is thought-provoking. By shifting the focus to individualistic alignment, the authors argue for AI systems that respect individual uniqueness, facilitating personalized AI development.

3. The VALUE INEQUITY INDEX (σINEQUITY) is a new metric for assessing the degree of impartiality in LMs' reasoning.

**Weaknesses:**

1. The reliance on WVS data, while innovative, may limit the applicability of results. Survey responses may not capture the full breadth of individual values, and the transformation of survey items into value-expressing statements could introduce biases or oversimplify complex beliefs.
2. The authors lack an analysis of the task's challenges and fail to sufficiently examine the reasons behind the poor performance of LLMs. Is the subpar performance primarily due to the complexity and contradictions in human preferences, or to the models' inadequate understanding of statements? What are the discrepancies between LLMs' CoT reasoning and users' actual preferences?
3. There are some minor errors in the paper. For example, in Figure 1, the correspondence between 1-10 and "satisfied" and "dissatisfied" on the left side seems to be reversed after data conversion. Is this an error in data processing, or is it only an issue with Figure 1?

**Questions:**

see weakness

---

### Note · Authors · 2024-12-15

I have read and agree with the venue's withdrawal policy on behalf of myself and my co-authors.